# Can achievement at medical admission tests predict future performance in postgraduate clinical assessments? A UK-based national cohort study

Lewis W Paton ,[1] I C McManus ,[2] Kevin Yet Fong Cheung ,[3] Daniel Thomas Smith ,[4] Paul A Tiffin [1,5]

[1]Department of Health Sciences, University of York, York, UK
[2]Research Department of Medical Education, UCL Medical School, London, UK
[3]Cambridge University Press & Assessment, University of Cambridge, Cambridge, UK
[4]General Medical Council, London, UK
[5]Health Professions Education Unit, Hull York Medical School, York, UK

**Correspondence to**
Dr Lewis W Paton;
lewis.paton@york.ac.uk

## ABSTRACT

**Objective** To determine whether scores on two undergraduate admissions tests (BioMedical Admissions Test (BMAT) and University Clinical Aptitude Test (UCAT)) predict performance on the postgraduate Membership of the Royal Colleges of Physicians (MRCP) examination, including the clinical examination Practical Assessment of Clinical Examination Skills (PACES).

**Design** National cohort study.

**Setting** Doctors who graduated medical school between 2006 and 2018.

**Participants** 3045 doctors who had sat BMAT, UCAT and the MRCP.

**Primary outcome measures** Passing each section of the MRCP at the first attempt, including the clinical assessment PACES.

**Results** Several BMAT and UCAT subtest scores displayed incremental predictive validity for performance on the first two (written) parts of the MRCP. Only *aptitude and skills* on BMAT (OR 1.34, 1.08 to 1.67, p=0.01) and *verbal reasoning* on UCAT (OR 1.34, 1.04 to 1.71, p=0.02) incrementally predicted passing PACES at the first attempt.

**Conclusions** Our results imply that the abilities assessed by *aptitude and skills* and *verbal reasoning* may be the most important cognitive attributes, of those routinely assessed at selection, for predicting future clinical performance. Selectors may wish to consider placing particular weight on scales assessing these attributes if they wish to select applicants likely to become more competent clinicians. These results are potentially relevant in an international context too, since many admission tests used globally, such as the Medical College Admission Test, assess similar abilities.

Traditionally, access to medical school has been based on educational attainment in secondary (high) school. Academic entry standards have partly been driven by strong competition for places as well as the intellectual demands of the courses. Moreover, prior educational attainment has been shown to predict performance in undergraduate and postgraduate medical training.[1–6] Applicants to medical school are also required to sit

## Strengths and limitations of this study

► We had access to a national sample of medical graduates.
► Restricting our analyses to those who had sat both BioMedical Admissions Test (BMAT) and University Clinical Aptitude Test (UCAT) allowed for a comparison of results across the two admissions tests.
► While our results are likely to be generalisable to all applicants who sit BMAT, it is unclear if this is the case for all applicants who sit UCAT.
► While a substantial proportion of doctors sits the Membership of the Royal Colleges of Physicians examination as part of their training, the generalisability of our results to other Royal College examinations is unclear.

additional standardised tests as part of the selection process. These have been introduced to help selectors further differentiate between educationally high-performing candidates and choose those considered to have the aptitude for a medical career. At times, it has also been hoped that such tests will facilitate widening access to medicine, if they are less sensitive to certain sociodemographic characteristics, compared with traditional metrics of attainment.[7]

The two main selection assessments currently used for entry into undergraduate medicine in UK medical schools are BioMedical Admissions Test (BMAT)[8] and University Clinical Aptitude Test (UCAT— previously known as UKCAT).[9] Both tests consist of a number of subtests,[10 11] described in table 1. These admissions tests are used in conjunction with other selection measures, such as interviews (including multiple mini-interviews), personal statements, references and educational attainment. Applicants to undergraduate medicine in the UK may apply to up to four different universities in a

**Table 1** Description of the three tests considered in this study, alongside a description of the subtests and what they aim to assess

| Test | Subtests |
|---|---|
| BioMedical Admissions Test[8 10] | **Aptitude and Skills**—'problem solving and understanding argument'. Replaced by **thinking skills** in 2020.<br><br>**Scientific Knowledge and Applications**—'the ability to apply scientific knowledge typically covered in school science and mathematics by the age of 16'.<br><br>**Writing**—'the ability to select, develop and organise ideas and to communicate them in writing, concisely and effectively'. |
| University Clinical Aptitude Test[9 11] | **Abstract Reasoning**—'assesses the use of convergent and divergent thinking to infer relationships from information'.<br><br>**Decision Making**—'assesses the ability to make sound decisions and judgements using complex information'. Prior to 2017, **decision analysis** was assessed rather than decision making.<br><br>**Quantitative Reasoning**—'assesses the ability to critically evaluate information presented in numerical form'.<br><br>**Verbal Reasoning**—'assesses the ability to critically evaluate information presented in written form'.<br><br>**Situational Judgement Test**—'measures the capacity to understand real-world situations and to identify critical factors and appropriate behaviour in dealing with them'. Introduced in 2013, not considered in this study. |
| Membership of the Royal Colleges of Physicians of the UK[18 19] | **Part 1**—assess 'a broad range of appropriate knowledge'.<br><br>**Part 2**—test the acquisition of 'a representative sample of medical knowledge, skills and behaviour'.<br><br>**Practical Assessment of Clinical Examination Skills**—designed to test 'clinical knowledge and skills', and involves candidates examining real patients with real clinical signs in various organ systems, as well as taking histories and carrying out clinical communication, usually with surrogate patients |

particular application cycle. As such, candidates may sit one or both of BMAT and UCAT, depending on the entry requirements of their chosen medical schools. These entry requirements may change year on year. Successful applicants to medical school then embark on an undergraduate medical course generally consisting of 5 years. Medical graduates then undertake the 2-year Foundation Programme, before embarking on specialty training in their chosen field of medicine. Each specialty is aligned with a particular Royal College, and as part of their specialty training, doctors must pass the Royal College membership examinations. Some of these examinations can be taken during the Foundation Programme prior to commencing specialty training.

For any test used in admission processes, it is important that scores from the test predict outcomes of interest. It is reasonably well established internationally that admission tests predict undergraduate medical education outcomes. In the case of BMAT, it has previously been reported that scores are predictive of first-year examination scores,[12] and that scores on *scientific knowledge and applications* correlate with examination marks more than scores on *aptitude and skills*. UCAT scores are predictive of performance throughout undergraduate medical school.[13–15] In the case of the UCAT, the scores have been shown to demonstrate incremental validity, above and beyond that provided by prior educational attainment, and additionally that scores on the *verbal reasoning* subtest had the strongest relationship with undergraduate academic achievement.[13] However, as medical training

extends beyond the initial undergraduate experience, it is important that we consider the predictive validity of admissions tests for relevant postgraduate outcomes. In this regard, there is relatively little published evidence. There are some indications that BMAT and UCAT scores are associated with performance in the written components of two UK Royal College membership examinations (equivalent to US board examination).[16 17] Thus, there is some emerging evidence that these assessments add value to the selection process when determining which applicants may be best suited to the academic challenges of early medical training. However, so that selectors have as much information as possible, it is important that further research into the predictive validity of BMAT and UCAT for postgraduate outcomes is carried out.

Furthermore, performance in written examinations is perhaps less important than how an individual behaves in actual clinical practise. While relevant semantic knowledge is vital in order to deliver safe and effective care, other skills, such as problem solving ability, are also likely to be crucial. Potential doctors are increasingly assessed on these abilities in a variety of ways, notably by selection assessments. Indeed, the admissions tests currently most widely used consist of a number of subsections, some of which aim to evaluate more fluid concepts of cognitive ability, in addition to sections evaluating the ability to recall factual knowledge.

However, for several reasons, linking performance on admissions assessments to aspects of actual clinical care and patient outcomes is extremely challenging. One

major barrier would be distinguishing between the impact of the individual doctor, versus the team or service, on a care outcome. Nevertheless, simulation-based examinations may serve as a plausible proxy for actual clinical behaviour, although reflecting 'maximal' rather than 'typical' performance. Such examinations are commonly taken as part of postgraduate medical training. One of the most widely taken Royal College examinations in the UK is the *Membership of the Royal Colleges of Physicians of the UK (MRCP(UK))*.[18] The MRCP(UK), referred to hereafter as the MRCP, consists of three parts,[19] including the *Practical Assessment of Clinical Examination Skills (PACES)* (see table 1). Trainees must complete all parts of the examination to advance to higher specialty training in the UK.

The creation of the UK Medical Education Database (UKMED)[20] has generated a longitudinal linked data set relating to individuals from application to medical school, through graduation and into post-qualification medical training. The UKMED has now matured to the point that there is a valuable opportunity to evaluate the extent to which performance at selection into medical school is reflected in performance in clinical assessments, many years later, following graduation. Specifically, it would be possible to evaluate which components of selection assessments have the most incremental predictive validity. In doing so, we draw from the 'individual differences' field, and, in particular, the psychometrics of intelligence, and how these can be applied to the interpretation of cognitive test scores, such as those assessed by BMAT and UCAT.[21] That is, what does the observed performance on different aspects of such assessments tell us about the likely underlying traits and abilities of the test-taker, relative to others?

This study, thus, had the following specific aims:

► To assess to what extent performance on the different components of BMAT and UCAT predicts performance on the MRCP examinations, with a particular focus on the clinical assessment PACES.

► To determine whether this predictive validity was incremental, over that provided by secondary (high) school educational attainment—the other main intellectual ability measure commonly used in the selection of medical students.

The findings would have clear implications for how scores from such tools are used within the selection process, and, in particular, the relative weight that should be placed on those components mainly testing semantic recall, compared with problem solving skills. In the context of UK medical selection, understanding the predictive profiles of BMAT and UCAT would help selectors choose an admission test they felt best suited their local requirements. However, as similar cognitive assessments are used globally in medical selection our results would have international relevance.

## METHODS

Note that all data and results presented in this paper are blunted in line with UKMED statistical disclosure controls.[22]

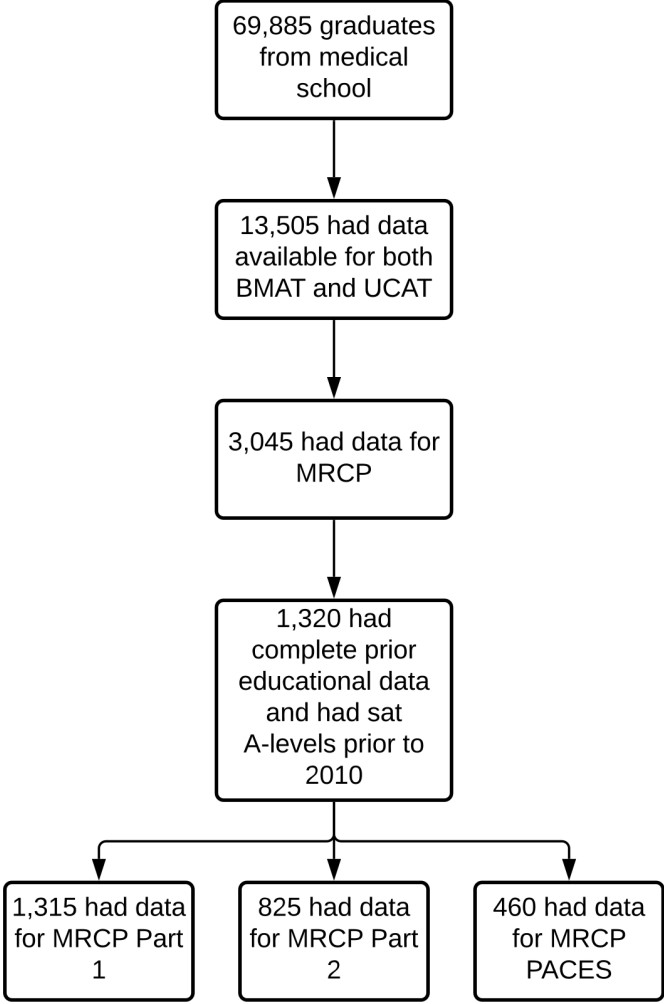

**Figure 1** The flow of data through this study. BMAT, BioMedical Admissions Test; MRCP, Membership of the Royal Colleges of Physicians; PACES, Practical Assessment of Clinical Examination Skills; UCAT, University Clinical Aptitude Test.

### Data availability and preparation

All data used in the analyses were obtained from the UKMED.[20] Data were available for 69 885 medical graduates between 2006 and 2018. Of these, 13 090 graduates had MRCP outcome data. We restricted analyses to those students who had sat both BMAT and UCAT; that is, 3045 graduates. This was to enable a comparison of the predictive validity between the two admissions tests. Figure 1 shows the flow of data through this study. It should be noted that the majority of graduates who had sat BMAT had also sat the UCAT, though the reverse was not true.

### Outcome variables

The primary outcomes of interest were the odds of passing each section of the MRCP at the first attempt. Performance at the first attempt at each section of the MRCP was chosen as this has been shown to be a good indicator of overall performance.[23] Predictions relating to secondary outcomes, the continuous 'score relative to pass at first sitting' for the examination components, were

also analysed. The results relating to these secondary outcomes are presented in online supplemental digital appendix 1.

### Predictor variables

In order to make our results relevant to selectors, performance at the most recent attempt at each section of both admissions tests was used, since it is these scores that are associated with admission to medical school. The subtest scores of BMAT (*aptitude and skills, scientific knowledge and applications* and *writing*) and the UCAT (*abstract reasoning, decision analysis, quantitative reasoning* and *verbal reasoning*) were standardised as z-scores (mean 0, SD 1) within each cohort of test takers (including unsuccessful applicants) and used as independent variables in our model. This standardisation process was intended to increase the comparability of selection assessment scores over time, as these are known to change.

As medical school selectors often base decisions on combined scores on the admissions tests, we also included a number of summed scales. We included summed BMAT *aptitude and skills* and *scientific knowledge and applications* score, denoted *total BMAT score*. We also included *total UCAT score* on all four scales. Additionally, it has been previously been shown that UCAT can be conceptualised as measuring two dimensions of cognitive functioning; verbal and non-verbal reasoning.[24] Thus, we also created a 'rebalanced' total UCAT score in order to adjust for the high relative weighting on non-verbal reasoning created by simply summing the scores from all four subtests. This 'rebalanced' score is the average of the three non-verbal scales (*abstract reasoning, decision analysis, quantitative reasoning*) added to the *verbal reasoning* score. All these summary scores were standardised within each cohort.

It was important to assess the incremental predictive validity of the two admissions tests, over and above that provided by conventional education achievement at secondary school. Therefore, two measures of prior academic attainment were used as covariates in the predictive models evaluated. First, A-level performance was used for those students from English secondary schools. As in previous research,[14] we used each entrant's best three A-level results, excluding 'General Studies'. We restricted analyses to those with A-levels sat prior to the introduction of A* grades in 2010. Achieved grades were awarded a tariff score (A=10, B=8, etc). Thus, the maximum tariff for each entrant was 30. The tariff scores were then standardised as z-scores by cohort of applicants. The second measure of prior academic attainment, we used was General Certificate of Secondary Education (GCSE) performance. Data were available on the sum of the nine best GCSE grades each candidate scored, where A*=6, A=5, B=4 and so on. 'Double awards' were counted as two separate GCSEs. Summed performance was standardised as z-scores by cohort of applicants. The dates of sitting of GCSEs were not available in the data. Thus, we determined cohorts by year of first sitting of an admissions test, assuming that the majority of candidates from the

same GCSE cohort will belong to the same admissions test cohort. This approach would not achieve perfect accuracy, since some individuals will choose to take a gap year or defer entry into university. However, it represented the optimum way, pragmatically, to estimate the year of GCSE sitting, in order to attempt to adjust for 'grade inflation'.

### Data analysis

Multilevel logistic regression models were built for the dichotomous outcome 'pass at first sitting', for each section of the MRCP. Multilevel models were used to allow the effects of the predictors to vary across medical school attended (via a random intercept), as previous research has shown that average MRCP scores are, to some extent, related to the particular medical school attended.[25]

Models were built for each subtest score of BMAT and the UCAT as well as the summary measures of the overall scores described above. Models were built that were both unadjusted and adjusted for prior educational attainment (GCSE and A-level performance). Only information from graduates with complete prior educational data was included in the models, to ensure full nesting of the models. Therefore, n=1320 individuals were included in our final analysis, including 1315 who had data for part 1 of the MRCP, 825 who had part two data and 460 who had PACES data. Of these, at the first attempt, 975 passed part 1 715 passed part 2 and 310 passed PACES.

### Missing data

Multiple imputation was used as a form of sensitivity analysis to assess the impact of missing prior educational data (GCSEs and A-levels) on our results. Standardised GCSE and A-level scores were imputed using multiple imputations by chained equations with linear regression from other educational data and sociodemographic variables. The number of imputations was chosen to ensure stability of results. We analysed the imputed data set in the manner described above and compared regression coefficients with the models performed on non-imputed data. This allowed us to assess the impact of missing educational data on our results.

All analyses were performed in the Health Informatics Centre Safe Haven using Stata V.14.[26]

### Patient and public involvement

None.

## RESULTS
### Descriptive statistics

Table 2 displays descriptive statistics for our cohort. Compared with the wider cohort of medical graduates, those who sat both BMAT and UCAT had higher academic attainment. These differences were all statistically significant at the p=0.05 level on Mann-Whitney U testing. Of those graduates who had sat both BMAT and UCAT, those who had also sat the MRCP had somewhat higher scores on the first two sections of BMAT, but lower scores on the

**Table 2** Descriptive statistics across various sub-cohorts of the dataset

| | All graduates (n=69 885 in total, including 20 175 individuals who sat BMAT and 43 685 individuals who sat UCAT) | Entrants who sat both BMAT and UCAT (n=13 505) | Individuals who sat BMAT, UCAT and had data available for at least one section of the MRCP (n=3045) | Individuals who sat BMAT, UCAT, MRCP, had complete prior educational data and sat A levels prior to 2010 (n=1320) |
|---|---|---|---|---|
| Age at BMAT | 17.73 (1.67) | 17.68 (1.47) | 17.62 (1.31) | 17.46 (0.92) |
| Age at UCAT | 18.59 (2.91) | 17.74 (1.54) | 17.70 (1.39) | 17.47 (0.94) |
| Sex | 43.02% men | 47.04% men | 48.05% men | 48.33% men |
| BMAT: AaS | 5.18 (1.06) | 5.19 (1.13) | 5.28 (1.15) | 5.28 (1.10) |
| BMAT: SKaA | 5.09 (1.00) | 5.13 (1.05) | 5.24 (1.08) | 5.25 (1.04) |
| BMAT: W | 8.92 (2.32) | 9.20 (2.30) | 8.51 (1.98) | 8.49 (1.89) |
| UCAT: AR | 636.76 (82.36) | 648.91 (81.13) | 644.42 (80.21) | 647.15 (79.41) |
| UCAT: DA | 659.44 (89.34) | 673.06 (86.54) | 666.82 (89.17) | 673.58 (88.52) |
| UCAT: QR | 668.82 (81.49) | 684.76 (80.86) | 677.42 (77.87) | 687.53 (70.17) |
| UCAT: VR | 617.81 (76.48) | 626.37 (76.05) | 635.07 (78.04) | 637.96 (73.33) |
| A-level performance, standardised by applicants | 0.04 (0.98) | 0.25 (0.79) | 0.27 (0.71) | 0.20 (0.73) |
| GCSE performance, standardised by applicants | 0.29 (0.83) | 0.47 (0.80) | 0.55 (0.76) | 0.65 (0.58) |

Means and SD are presented, except for *sex* where percentages are displayed.

AaS, aptitude and skills; AR, abstract reasoning; BMAT, BioMedical Admissions Test; DA, decision analysis; QR, quantitative reasoning; SKaA, scientific knowledge and applications; UCAT, University Clinical Aptitude Test; VR, verbal reasoning; W, writing.

writing component of BMAT as well as each section of the UCAT. Again, these differences were statistically significant at the p=0.05 level on formal testing.

### Predictive validity of BMAT and UCAT for MRCP performance

Figure 2 displays the predictive and incremental validity for BMAT subtest and summary scores for passing each section of the MRCP at the first attempt. Figure 3 displays the same values for the UCAT subtest and summary scores. In both figures, the circle represents the odds ratio (OR) from the univariable (or unadjusted) multilevel logistic regression models. The triangle represents the OR from the multivariable (or adjusted) models. Tables displaying these results are available in online supplemental digital appendix 1.

As is seen in figure 2, passing the first part of the MRCP is incrementally predicted by scores on *aptitude and skills* (OR 1.22, 1.04 to 1.44, p=0.02) and scores on *scientific knowledge and applications* (OR 1.49, 1.25 to 1.78, p<0.001). This later result can be interpreted as follows. For every SD above the mean an individual scored on *scientific knowledge and applications,* their odds of passing part 1 of the MRCP at the first attempt increased by around 49%. Similarly, performance on both *aptitude and skills* (OR 1.36, 1.05 to 1.75, p=0.02) and *scientific knowledge and applications* (OR 1.35, 1.04 to 1.75, p=0.02) incrementally predicts passing part 2 of the MRCP at the first attempt. When it comes to passing the clinical assessment PACES, evidence of

predictive validity was only observed for the *aptitude and skills* score (OR 1.34, 1.08 to 1.67, p=0.01), and this result is independent of prior educational attainment. 'Total' BMAT score (ie, total score across the first two sections) is a significant predictor of passing all sections of the MRCP at the first attempt. In contrast, performance on the writing component of BMAT is not a significant predictor for any of the outcome variables analysed.

In the case of UCAT (figure 3), *quantitative reasoning* (OR 1.37, 1.16 to 1.62, p<0.001) and *verbal reasoning* (OR 1.24, 1.05 to 1.48, p=0.01) scores incrementally predicted passing the first, knowledge-based, part of the MRCP at the first attempt. Higher *verbal reasoning* scores (OR 1.55, 1.17 to 2.04, p<0.01) are associated with a higher odds of passing part 2 (applied knowledge) of the MRCP at the first attempt. However, a higher score on the UCAT *abstract reasoning* subtest was associated with *lower* odds of passing part 2 of the MRCP examination at the first attempt, independently of prior educational attainment (OR 0.79, 0.64 to 0.99, p=0.04). Only the UCAT *verbal reasoning* scores were statistically significantly associated with the odds of passing the clinical assessment, PACES, at first attempt (OR 1.34, 1.04 to 1.71, p=0.02). In this case, the odds of passing the PACES at first attempt increased by around a third for every SD above the mean achieved on the UCAT *verbal reasoning* subtest, for applicants sitting the test. The total UCAT score is a significant

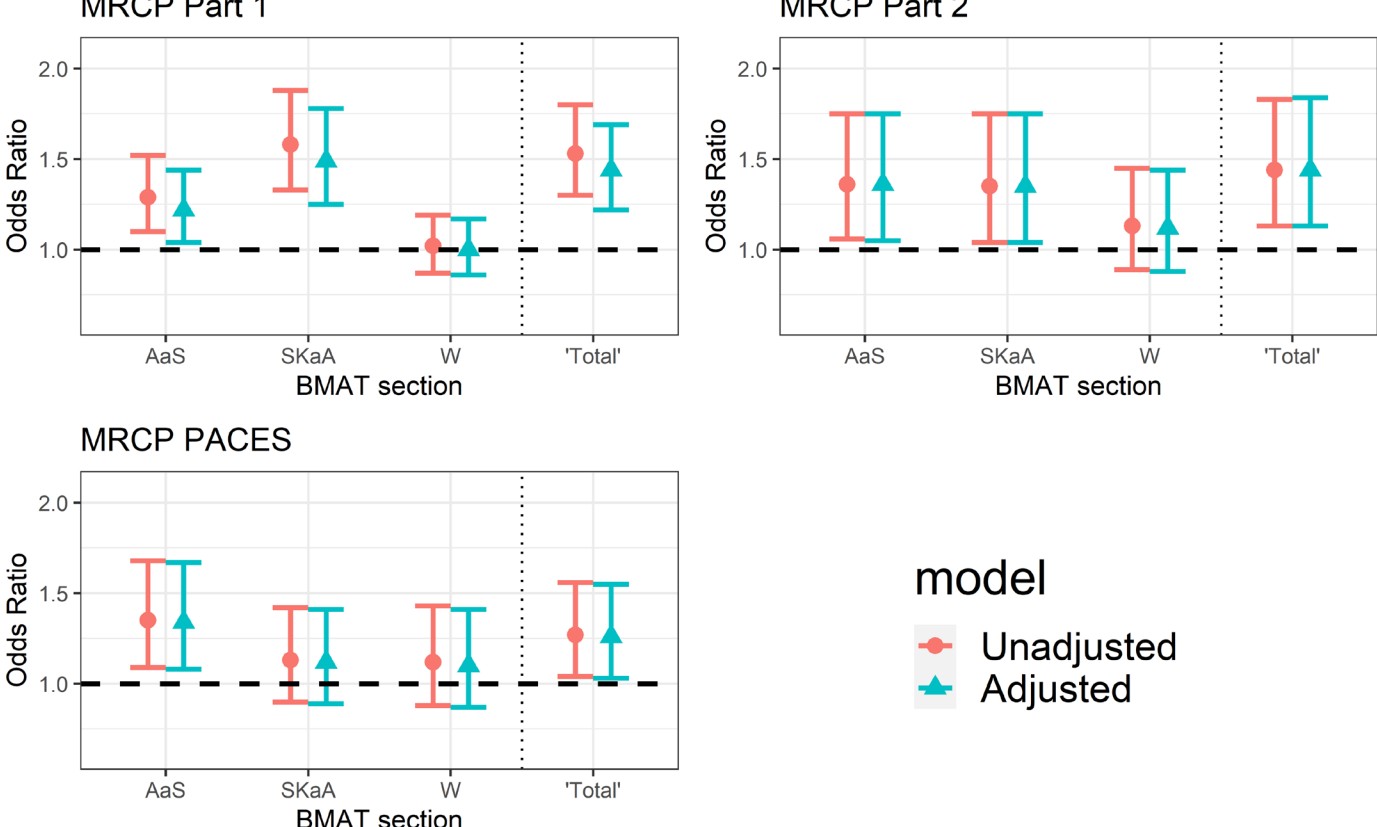

**Figure 2** Predictive validity of each standardised section score of BMAT for passing each section of the MRCP at the first attempt. Results are shown both unadjusted and adjusted for prior educational performance. AaS, aptitude and skills; BMAT, BioMedical Admissions Test; MRCP, Membership of the Royal Colleges of Physicians; PACES, Practical Assessment of Clinical Examination Skills; SKaA, scientific knowledge and applications; W, writing task.

independent predictor of passing part 1 of the MRCP, but not subsequent sections of the exam. However, the 'rebalanced' UCAT score, which weights verbal and non-verbal performance equally, is a statistically significant independent predictor of passing all sections of the MRCP, including PACES (OR 1.34, 1.03 to 1.75, p=0.03). Indeed, the odds of passing the PACES at first attempt increase by around a third for every SD above the mean achieved for applicants sitting the test. No statistically significant associations with the outcomes of interest and the UCAT *decision analysis* scores were observed.

### Imputation of prior educational attainment

In those who sat both BMAT and UCAT, missing values of A-levels and GCSE scores were imputed. We used chained imputation methods, building models on available sociodemographic variables and educational variables, where present. Imputation results stabilised after 20 iterations. Logistic regression models, controlling for the influence of A-levels and GCSEs, were fitted to the imputed data set to assess the incremental predictive validity of each scale score for predicting whether an individual passed each section of the MRCP on first attempt. As a form of sensitivity analysis, regression coefficients were compared with coefficients obtained when fitting the same model to the non-imputed data (as presented in figures 2 and 3).

Full results from models on imputed data are available in online supplemental digital appendix 1.

In general, the results from models fitted to imputed data are broadly similar to those fitted to non-imputed data. Only small differences in effect size or p values were observed when predicting performance on part 1 of the MRCP. No overall changes in statistical significance were observed. For part 2 of the MRCP, no differences were observed in the predictive validity of BMAT subtest scores. There are some differences in how the UCAT subtest scores behave, however. The a*bstract reasoning* subtest score becomes a non-statistically significant predictor, and the *quantitative reasoning* and total UCAT become statistically significant predictors (p<0.05). For predicting the performance at PACES, the UCAT *decision analysis* score becomes a predictor of only borderline statistical significance, and the total UCAT score becomes statistically significant. Again, negligible differences were observed in relation to the predictive validity of BMAT subtest scores.

### DISCUSSION

In this study, we observed that both BMAT and UCAT scores have incremental predictive validity for performance on the MRCP. As might be expected, scores on different sections of the admissions tests predict

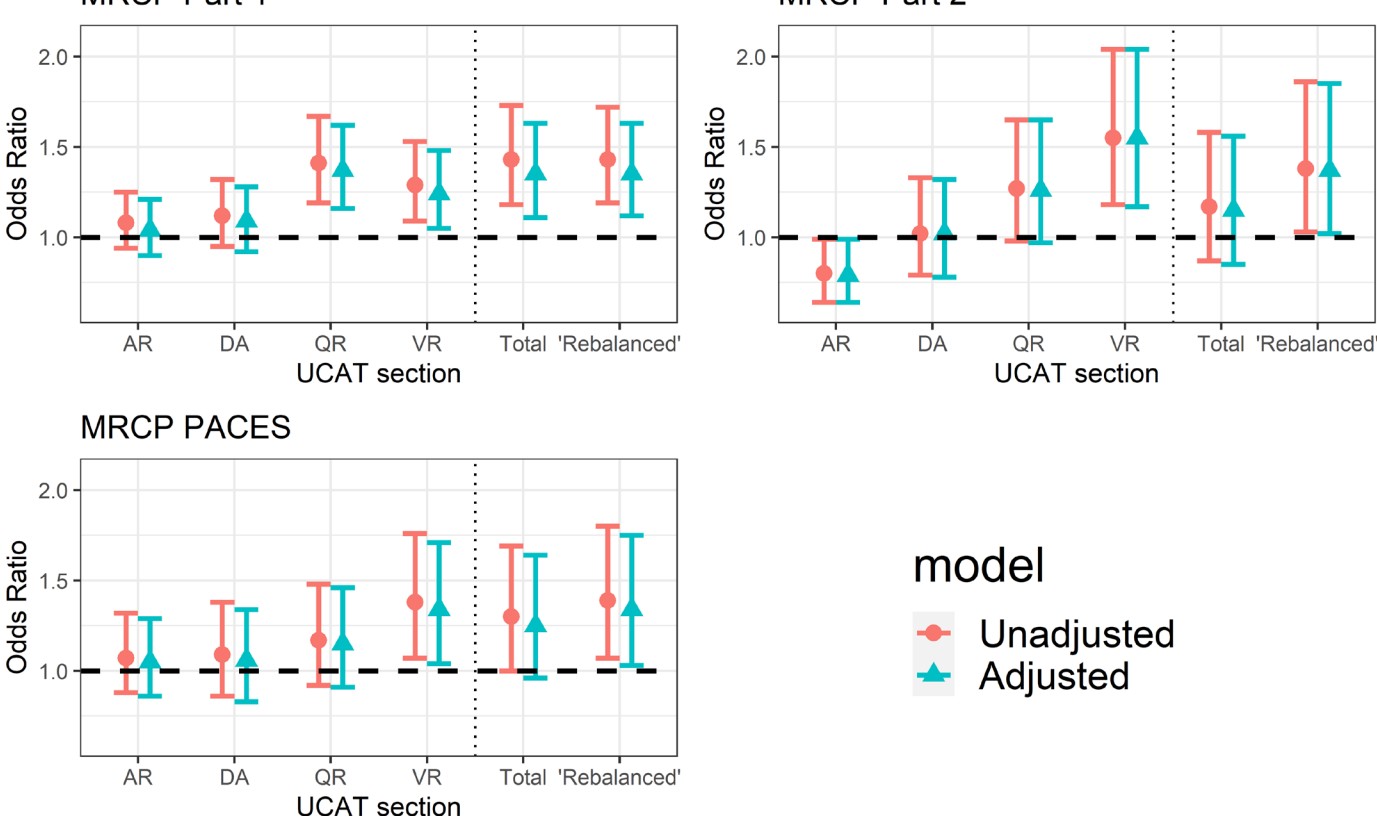

**Figure 3** Predictive validity of each standardised section score of the UCAT for passing each section of the MRCP at the first attempt. Results are shown both unadjusted and adjusted for prior educational performance. AR, abstract reasoning; DA, decision analysis; MRCP, Membership of the Royal Colleges of Physicians; QR, quantitative reasoning; UCAT, University Clinical Aptitude Test; VR, verbal reasoning.

performance on the various sections of the MRCP. Performance on the BMAT *aptitude and skills* subtest and on the UCAT *verbal reasoning* subtest predicted the odds of passing all sections of the MRCP at first attempt, independently of prior educational attainment.

Of particular interest was the relationship between the selection assessment scores and performance on the PACES, as an assessment of practical clinical skills. In those graduates who had originally sat both BMAT and the UCAT, we observed that scores on BMAT *aptitude and skills* (OR 1.34, 1.08 to 1.67) and the UCAT *verbal reasoning* (OR 1.34, 1.04 to 1.71) subtests were incrementally predictive of passing the PACES at the first attempt. We also noted that the summed scores, across the first two BMAT subtests, predicted performance on all sections of the MRCP. It was also observed that a 'rebalanced' UCAT summary score, which addresses the substantial weight on non-verbal reasoning, appears to add value over using the usual summary, total UCAT score. This was especially evident when predicting performance on the PACES, where only the 'rebalanced' total was incrementally predictive.

Consistently, BMAT *aptitude and skills* and UCAT *verbal reasoning* subtest scores showed predictive validity when considering all the MRCP-related outcomes of interest. This is in line with findings from a previous study of the

predictive validity of the UCAT scores, where performance on the *verbal reasoning* subtest had the strongest relationship with academic achievement throughout medical school study.[13] As with the present findings, this predictive ability was incremental over and above that provided by prior educational attainment at secondary (high) school. Thus, it is reasonable to infer that these assessment sections evaluate abilities that are both relevant to future performance in clinical tests of both pure and applied semantic knowledge and procedural skills. Moreover, at least, to some extent, they must measure traits or abilities not fully covered by the, largely science based, subjects taken at secondary school, especially at A-Level. Previous research has shown that the BMAT *aptitude and skills* subtest can be conceptualised as measuring 'thinking skills',[27] that is, problem solving ability. The UCAT *verbal reasoning* subtest, as mentioned earlier, evaluates the ability to make inferences and draw conclusions from written information and can be considered related, though somewhat distinct from, problem solving that does not heavily rely on verbal comprehension skills.[24] Thus, both selection assessment subtests test cognitive, problem solving skills, rather than semantic, factual recall, though verbal reasoning does not seek to assess such non-verbal skills. Nevertheless, both subtests generated scores with a similar ability to predict subsequent performance in PACES.

Our findings can be considered alongside results from a study assessing the relationship between BMAT and UCAT scores and performance on the Membership of the Royal College of Surgeons (MRCS) examination.[17] In this instance, using data from a comparable time period, the authors observed a statistically significant relationship between admissions test scores and performance on the written component of the MRCS, and evidence of *aptitude and skills* and *abstract reasoning* scores incrementally predicting performance on the clinical examination component. It is not reported how this study of MRCS performance controlled for the fact that some candidates had sat one of the admissions tests, while others had sat both. Thus, comparisons across the two admissions tests within that study, and indeed direct comparisons with this study, are difficult to make. Nevertheless, taken together, it appears there is some emerging evidence regarding the relationship between admissions test scores and performance on postgraduate clinical assessments more broadly. In both this study and the MRCS study, it is the scores on BMAT *aptitude and skills* that predicts performance at the Royal College clinical assessment, rather than scores on the component of BMAT, which assesses more factual recall. The UCAT scale scores which possess predictive power for the clinical assessment differs between the MRCP, where *verbal reasoning* scores are statistically significant, and the MRCS, where it is the *abstract reasoning* scores. Such a difference may by explained simply by the different skills required to pass a surgical clinical assessment and those needed for a more general internal medicine clinical assessment. However, taking all the results together, it suggests that it is indeed cognitive, problem solving skills, which are particularly important for predicting performance at Royal College clinical assessments, rather than semantic, factual recall.

There are some advanced school qualifications, such as 'thinking skills'[28] that may evaluate such cognitive, problem solving skills, though they are not generally accepted as part of entry requirements by UK medical schools. However, it may be that the high school maths and science exams, generally taken by medical school applicants, do not always test these abilities fully, typically placing more emphasis on testing non-verbal problem solving skills and semantic factual recall. Also, as previously illustrated via modelling studies, academic or educational performance is not simply a direct result of intellectual (cognitive) ability but can mediated by separate factors such as secondary school attended[29] or personality traits such as conscientiousness.[30 31] Indeed, more recently, there have been attempts to identify the genetic basis of 'non-cognitive' abilities by partitioning education achievement from intellectual ability.[32] Moreover, when predicting performance at 'high fidelity' clinical assessments, abilities related to interpersonal functioning, may be stronger predictors than levels of relevant semantic knowledge.[33] Within the individual differences tradition, such traits and abilities are often conceptualised within an 'emotional intelligence' framework.[34] However, the

distinction between such 'non-cognitive' and 'cognitive' abilities is not always well demarcated.[35] This is especially interesting in the case of verbal ability. Indeed, the earliest attempts to measure 'social intelligence' found it difficult to delineate between verbal ability and experimental measures of interpersonal competence.[36] Individuals are often judged in regards to their 'social skills' on the basis of the language they use, as much as their behaviour. Thus, in this case, to some extent, verbal reasoning, though conventionally conceptualised as a 'cognitive ability', could still be an indicator of interpersonal skills that would be important in performing well in tests that involved simulated patients or clinical situations.

A counterintuitive result observed was that for a higher UCAT *abstract reasoning* score to be associated with a lower odds of passing MRCP part 2 at the first attempt. However, once missing prior educational data were imputed, the abstract reasoning scores became a non-statistically significant predictor of MRCP part 2 performance. This suggests that caution should be used when interpreting the former result. Additionally, scores on the UCAT *decision analysis* subtest, as well as UCAT total score, became significant predictors of PACES performance in the imputed dataset.

Adjusting for A-level performance alone had very little impact on the results. This is why only the findings adjusted for both A-level and GCSE performance were presented. This is almost certainly due to the lack of variation in A-level performance for the final sample of doctors included in the analysis. For example, over 70% of medical graduates with A-level data from before 2010 achieved the maximum available grade of AAA. This lack of observed variation would have been further constrained in the final sample of doctors sitting the MRCP relatively early in their careers, who may have been particularly academically well performing. Adjusting for GCSE performance, where the examinations had been taken around the age of 16 years, had a somewhat greater impact on the adjusted results, particularly when predicting performance on part 1 of the MRCP. This increased impact of adjusting for GCSE, rather than A-level performance, is likely due to greater variation in grades across the cohort. For future analyses, it should be noted that, from 2010 onwards, the introduction of an A* grade at A-levels will have increased the variability of the grades obtained by medical students and doctors.

### Strengths and potential limitations
We restricted our analytic sample to those individuals who sat both BMAT and UCAT. This allowed for comparison of results across the two admissions tests. However, by restricting the analysis in this way, there is a risk that our results may not generalise to those individuals who sat just one of the admissions tests. As could be seen, the cohort analysed here had generally higher academic attainment than the wider cohort of medical graduates available within the UKMED. In practice, due to the relatively low proportion of medical schools using BMAT, compared

with the UCAT in the UK, currently relatively few medical applicants in the UK sit only BMAT, but a much greater number of applicants sit only the UCAT. As such, our results are likely to be generalisable to the wider cohort of all applicants who sit BMAT. However, the extent to which our findings would generalise to all applicants who sit the UCAT is uncertain. A number of non-statistically significant trends were noted between selection assessment scores and performance on membership exam components. Due to the stage of maturity of the cohorts being followed via the UKMED, the numbers of individuals having sat the MRCP at least once were relatively low. This may mean that the study was underpowered to detect relatively small effect sizes. Nevertheless, it is doubtful whether such modest effects would be of 'educational significance', even if they were found to be statistically significant. However, our analyses could be replicated at a later stage when more data are available relating to postgraduate clinical examination performance with the UKMED.

The generalisability of these results to other Royal College examinations is also unclear. Although a substantial proportion of doctors sits the MRCP as part of medical training, they may not be entirely representative of all those in postgraduate training. However, broadly similar results have been observed in relation to performance on the MRCS examination[17] as observed in this study, although, as discussed earlier, caution should be exercised in making direct comparisons between the studies. As more data become available via the UKMED, the presence of similar relationships between selection test scores and other Royal College examinations could be evaluated for, and as UKMED continues to mature, a sample, including those who sat the MRCP relatively later in their training could additionally be evaluated.

The selection assessment scores were standardised using cohort means and SD. This was the most appropriate method of including scores in analyses across years. However, some limitations of this approach should be acknowledged. The approach assumes that the ability of the population of test takers (on any test construct) is stable year on year. This may not always be the case, partly as the schools which required each test for admission has changed over time. However, such an adjustment was necessary as non-standardised scores would be rendered even less comparable across cohorts. For example, secular trends may be at work. As selection tests have become widely used and established, applicants may have spent more time developing the skills tested by these examinations, so later cohorts are likely to have higher ability, on average, compared with previous ones. Our approach to standardisation may have helped address such temporal influences.

Finally, the rationale for our study mainly rests on the premise that postgraduate clinical assessments may be a proxy for real-world performance. Ideally, admission scores would be linked to metrics of actual clinical performance. However, this is extremely challenging. It may be considered that competent performance in a clinical assessment is perhaps a necessary, though not sufficient, condition for actual workplace practise. In this regard, it has been reported that in non-UK medical graduates, increased performance in the practical component of a licensing examination was protective of the risk of future fitness to practise events.[37]

## Potential policy implications

When considering evidence for the validity of an assessment, it is sometimes useful to apply Kane's validity framework.[38] This is an argument-based approach, whereby a critique of an assessment is placed in the context of the claims made for the test as well as how the resulting test scores are used in practice. In this regard, the claims made for the UCAT are more ambitious than those of the developers of BMAT. Specifically, the UCAT developers claim that the assessment is *'seeking to identify the characteristics in applicants which will make them good clinicians and thus to improve the quality of those who enter the professions with the ultimate aim of improving patient care'*.[39] Our findings here suggest that the UCAT could potentially be useful in identifying medical applicants likely to become good future clinicians, if we accept the assumption that MRCP performances, and, in particular, performance on PACES, are a reasonable proxy for actual clinical behaviour. Testing this assumption is, as mentioned earlier, non-trivial. However, there is evidence that clinical examinations have face validity with clinicians,[40] which suggests this assumption may indeed be plausible.

Nevertheless, the potential of the UCAT to improve medical selection with regards to selecting those applicants likely to make good doctors will not be fully realised unless the resultant scores are reported and used by selectors in the optimally effective way. In particular, it should be noted that reweighting the summary UCAT scores into a 'rebalanced' UCAT summary score, which did not overemphasise non-verbal reasoning, appeared to provide greater effect sizes, particularly when predicting performance on part 2 and the PACES component of the MRCP. Selectors themselves may also wish to consider placing more weight on the verbal reasoning component of the UCAT if they wish to realise additional incremental predictive ability for both undergraduate and postgraduate educational performance. However, some caution should be exercised with this approach; previously it was reported that verbal reasoning was the UCAT cognitive subtest score that was most sensitive to socioeconomic status in medical school applicants.[7]

In contrast to the UCAT, BMAT claims, less ambitiously, that the assessment is intended to identify those with '… *potential to succeed on medical and health-related courses'*.[8] In this sense, the present findings largely support this claim, though performance on the writing task section seemed to have little relationship with postgraduate performance in this sample. Moreover, despite the modest claimed aspirations, BMAT component scores and summed scores did seem to have some incremental predictive

ability regarding future PACES performance. Unsurprisingly, *scientific knowledge and application* scores tended to predict performance on part 1 and 2 of the MRCP, which depends on the ability to recall factual information. However, these predictions remained fairly substantial even after correcting for the influence of prior educational attainment. This suggests that such components of selection assessments may add value above secondary school qualifications, probably by providing a more fine-grained metric, which is able to differentiate test takers at the top end of ability. This is in contrast to the UK-based school examinations, such as A-levels and GCSEs, where many medical school applicants often achieve straight As (or, from 2010 onwards, A*s). Furthermore, such semantic knowledge tests may be of practical use in selection when considering applications from overseas candidates, where the equivalence of educational qualifications in the sciences may be uncertain, or difficult to equate. Nevertheless, consideration should be given to the potential impact on the diversity and widening access agenda if knowledge tests are to be included in the selection process. Such tests are likely to be more sensitive to sociodemographic background factors and coaching effects compared with measures of problem solving ability. However, it should be noted that, assessments of the latter are also prone, to some extent, to practice and coaching effects.[41] Moreover, if the overall goal of medical schools is to produce clinically competent doctors, rather than students and trainees who merely do well at examinations requiring factual recall, then more emphasis on the scores from assessments evaluating problem solving ability may be advisable.

It is important to note that, in this study, we consider only how the cognitive aspects of intelligence, as measured by BMAT and UCAT, relate to a proxy for clinical competence. However, producing clinically competent doctors is a complex multifaceted process. Prospective doctors are increasingly also assessed on 'non-cognitive' skills such as attitudes and knowledge of professionalism.[42] In order to select those most likely to become competent doctors, it is likely that selectors should also consider the evidence, if it exists, of how these 'non-cognitive' abilities relate to clinical competence too.

The use of such selection assessments, which often have components evaluating both semantic knowledge, and problem solving skills are increasingly being used globally. Consequently, our findings have implications for medical selection internationally. That scores from both BMAT and UCAT can predict such distal outcomes add to the international evidence supporting the use of admissions testing. Furthermore, those sections which assess problem solving or verbal reasoning added most value to the selection process for clinical assessment performance. As many admissions tests in use around the world assess such abilities, such as, for example, the Medical College Admissions Test used in North America, our findings may have implications for medical selection internationally. That being said, research should be undertaken

to determine if this indeed is the case in each context locally.

Closer to home, in the UK, undergraduate medical admission teams are generally faced with a choice of either BMAT or UCAT as the selection assessment to employ. The present findings do not highlight any clear advantage, in terms of predictive validity, of one over the other. Therefore, choices should be informed by other test qualities, such as the potential impact on widening access and diversifying medicine. In this sense, both BMAT and UCAT have components where performance is associated with certain candidate background factors.[7 43] Thus, at present, in this respect, there would seem to be no overall clear advantage of one selection assessment over the other.

Beyond selection, highlighting those aspects of cognitive intelligence which predict postgraduate clinical simulation performance, raises the possibility of targeted interventions for those who score less well on the relevant sections of BMAT or UCAT. However, further research would be required, both in the identification of students for intervention (eg, there may be sociodemographic factors to consider, which have not been accounted for in this selection-focussed study), and in the timing and nature of the intervention itself.

### Directions for future research

In addition to replicating these findings in other postgraduate examinations and subsequent cohorts of medical graduates, there are future opportunities, provided by UKMED, to evaluate the validity of medical selection assessments. In particular, a Situational Judgement Test (SJT) was added to the UCAT in 2013. Previous research has shown that scores on the SJT do possess some predictive validity for undergraduate supervisor ratings.[44] In this context, it will be important to consider to what extent the SJT scores predict performance in clinical assessments, and other construct-relevant outcomes, and whether this is incremental to other selection measures.

Another change made to the structure of the UCAT has been the replacement of the *decision analysis* subtest for *decision making* in 2017. Once the data availability allows, it would be important to estimate the relationship between *decision making* and the performance on PACES and other postgraduate practical examinations.

Given that the two selection assessments considered here had comparable predictive validity, as mentioned above, other properties should be considered when choosing such a test for use in medical admission processes. It has been highlighted that the trade-off between selecting for future performance and 'adverse impact' on certain under-represented groups can be conceptualised, and modelled, as a problem of 'pareto-optimisation'.[45] That is, there are optimal, and suboptimal, combinations of how elements within a selection process can be weighted and combined, with the goal of selecting the best doctors while maintaining or improving the diversity of the

medical workforce. Such information is vital if selectors are to make informed choices.

Ultimately, linking selection test scores to relevant patient outcomes in clinical practice would be the most desirable source of validity evidence. This would shed light on how, if at all, such assessment scores could predict typical performance, in contrast to the maximal performance observed in high stakes postgraduate clinical examinations. Nevertheless, as emphasised earlier, this is extremely challenging, though more advanced statistical techniques may eventually be able to disentangle environment from individual physician effects. Such approaches would depend on clinicians working across different (ideally small) teams in the hope that the individual doctor's clinical 'footprint' might be observable.

## CONCLUSIONS

Our findings suggest that both BMAT and UCAT contain components that show some incremental predictive validity for MRCP performance, with broadly similar effect sizes. The subtests *aptitude and skills* and *verbal reasoning* were of most value, over and above conventional metrics of educational performance, in this regard. Thus, it could be argued it is these aspects of cognitive performance, rather than factual recall, that may be of most value in the selection of future doctors in this context. Both test developers and selectors should consider this evidence when considering how such assessments should be constructed and used within the medical selection process.

**Acknowledgements** We would like to thank Rachel Greatrix at UCAT for her comments on an earlier version of this manuscript.

**Contributors** PAT led on conception of the project, with support from LWP, ICM, KYFC and DTS. LWP led on data analysis with support from PAT. All authors contributed to interpretation of the results. LWP led on writing of the manuscript, with support from ICM, KYFC, DTS and PAT. All authors have approved the final version of the article submitted. LWP is the guarantor of the paper.

**Funding** LWP is partly funded by the UCAT Consortium. The UCAT consortium partly funded this research but did not play an active role in determining the study design or reporting the results. University College London receives money from MRCP(UK) for the contribution of ICM to the running of the examination, but ICM receives none of that money. DTS is employed by the GMC as a data analyst working on the UKMED project. The views expressed here are his views and not the views of the GMC. PAT's research time for this project was funded by an NIHR Career Development Fellowship (CDF-2015-08-11). This paper presents independent research funded by the National Institute for Health Research (NIHR). The views expressed are those of the authors and not necessarily those of the NHS, the NIHR or the Department of Health and Social Care.

**Competing interests** UCAT pay for a portion of LWP's research time. LWP and PAT have received travel expenses for attendance at UCAT consortium meetings. PAT has previously received research funding from the UCAT consortium. KYFC is an employee of Cambridge Assessment—a group of exam boards that owns and administers the BioMedical Admissions Test (BMAT); UK GCSEs and A-levels; and International GCSEs and A-levels. ICM is member of various MRCP(UK) committees overseeing and analysing results from MRCP(UK) examinations.

**Patient consent for publication** Not applicable.

**Ethics approval** This study does not involve human participants.

**Provenance and peer review** Not commissioned; externally peer reviewed.

**Data availability statement** Data are available upon reasonable request. Data may be obtained from a third party and are not publicly available. The dataset supporting the conclusions of this article is available from UKMED on application (www.ukmed.ac.uk). UK Medical Education Database ('UKMED') P051 extract generated on 13 May 2019. Approved for publication on 20 November 2019. We are grateful to UKMED for the use of these data. However, UKMED bears no responsibility for their analysis or interpretation. The data include information derived from that collected by the Higher Education Statistics Agency Limited ('HESA') and provided to the GMC ('HESA Data'). Source: HESA Student Record 2002/03 to 2017/18 Copyright Higher Education Statistics Agency Limited. The Higher Education Statistics Agency Limited makes no warranty as to the accuracy of the HESA Data, cannot accept responsibility for any inferences or conclusions derived by third parties from data or other information supplied by it.

**ORCID iDs**
Lewis W Paton http://orcid.org/0000-0002-3328-5634
I C McManus http://orcid.org/0000-0003-3510-4814
Kevin Yet Fong Cheung http://orcid.org/0000-0002-9548-2932
Daniel Thomas Smith http://orcid.org/0000-0003-1215-5811
Paul A Tiffin http://orcid.org/0000-0003-1770-5034

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
