## [Reviewer comments · BMJ Open]

ARTICLE DETAILS

TITLE (PROVISIONAL)	Can achievement at medical admissions tests predict future performance in postgraduate clinical assessments? A UK-based national cohort study
AUTHORS	Paton, Lewis; McManus, Ian; Cheung, Kevin; Smith, Daniel; Tiffin, Paul

VERSION 1 – REVIEW

REVIEWER	Fabrizio Consorti University of Rome La Sapienza
REVIEW RETURNED	28-Oct-2021

GENERAL COMMENTS	The manuscript reports about a nationwide cohort study on the predictive value of the scores of some admission tests for the results of a postgraduate test, as a proxy of real clinical skill. The text is clear and all the relevant information are easily available to the reader (context, objectives, methods, results). I have only a minor issue with the Introduction, that is very long and could be shortened, omitting some of the detailed information about the BMAT, UCAT and MRCP. I suggest that the content of page 5, 6 and 7 could be summarized in one page and half, for the benefit of the reader, adding references (also web sites) for those who are interested in the details. The definitions of the constructs (aptitude and skills, ...) are very important and they could be reported in a separate table, itemized. More synthesis will help the reader to appreciate the lack of knowledge which motivated the study ("Currently, no evidence relates to how BMAT or UCAT scores relate to performance on PACES."), the goal and objectives of the study (page 8). A second point - a suggestion - is a clear statement on the theoretical perspective adopted by the authors. Selection and prediction are complex processes, based on assessment (see eg. Patterson, F., et al (2016). How effective are selection methods in medical education? A systematic review. Medical education, 50(1), 36–60.). Assessment is inherently based on values and theoretic assumptions must be declared, because the constructs that underly BMAT, UCAT are meaningful only in a theoretical framework. As far I can understand, the authors adopted a cognitive quantitative approach. A licit choice, that is not the only possible one. It could be confusing to mix up socio-cultural determinants or "personality traits such as conscientiousness ... inter-personal functioning ... social intelligence" (page 19) that belong to other theoretical backgrounds. I think it would be fair to declare that the study considers only some of the cognitive elements of the very complex process of the development of a "good doctor".
--

	Finally, another possible use of the results of the study could be to inform the educational process. Beyond using the scores to select students, a faculty could use the score to personalize some interventions to achieve the best possible outcome for every student.
--	---

REVIEWER	Reinaldo Bestetti University of Sao Paulo Faculty of Medicine of Ribeirao Preto, Medicine
REVIEW RETURNED	28-Oct-2021

GENERAL COMMENTS	n this paper, the authors have evaluated the role of Biomedical Admission test (BAMT) and University Clinical Admission Test (UCAT) in the prediction of academic achievement at the Membership of the Royal College of Physicians (MRCP) at the first attempt. The MRCP, a postgraduate assessment, is comprised of part I (knowledge-based), part II (applied knowledge), and part III - Practice Assessment of Clinical examination Skills (PACES), which is devoted to the assessment of aptitude and skills. A particular attention was paid to PACES. The authors studied 3045 subjects, and observed that the aptitude and skills as well as scientific knowledge and applications subtest of the BMAT predicted passing in MRCP I and II at the first attempt. However, aptitude and skills were the only subtests to predict passing in MRCP-PACES at the first attempt. Regarding UCAT, quantitative reasoning and verbal reasoning predicted passing in the MCRP part I, and verbal reasoning (the ability to problem-solving) predicted passing MCRP Part II and MCRP PACES. Importantly, these results were obtained independently of previous academic attainment at high school. Thus, they concluded that “selectors may wish to consider placing particular weight on scales assessing these attributes “[“thinking skills”i.e problem solving ability -aptitude and skills) or verbal reasoning] if they wish to select applicants likely to become more competent clinicians.” GENERAL COMMENTS. The selection process of students to enter a medical school is a difficult task. The future doctor should acquire cognitive abilities, procedural skills, and compassion attitudes, which are difficult to predict at the entrance examination test. Therefore, it was reassuring to take a look at this paper in which we can see that the assessment of aptitude and skills and verbal reasoning, ultimately the capacity of problem-solving, are predictive of academic achievement at a postgraduate medical examination. SPECIFIC COMMENTS. By and large, the paper is nicely written. I could suggest some changes on it; 1) I think that the Introduction section is too long, and should be shortened accordingly. 2) In my view, the use of the term “thinking skills” is not appropriate. Therefore, I would suggest to the authors to keep the terms aptitude and skills instead; 3) As I understood the situation, attitudes were not assessed in BAMT, UCAT, and MCRP. If this is true, I think that it is an important limitation because attitudes are paramount for a good doctor; therefore, a few comments on the Discussion section would be desirable; 4) The Discussion section is also too long, and authors should focus on the explanation given to the results obtained. Therefore, I strongly suggest that the Discussion section be shortened.
---

VERSION 1 – AUTHOR RESPONSE

Reviewer: 1

Dr. Fabrizio Consorti, University of Rome La Sapienza

Comments to the Author:

The manuscript reports about a nationwide cohort study on the predictive value of the scores of some admission tests for the results of a postgraduate test, as a proxy of real clinical skill.

The text is clear and all the relevant information are easily available to the reader (context, objectives, methods, results).

Authors response: Thank you for your positive comments.

Reviewer 1: I have only a minor issue with the Introduction, that is very long and could be shortened, omitting some of the detailed information about the BMAT, UCAT and MRCP. I suggest that the content of page 5, 6 and 7 could be summarized in one page and half, for the benefit of the reader, adding references (also web sites) for those who are interested in the details. The definitions of the constructs (aptitude and skills, ...) are very important and they could be reported in a separate table, itemized.

Authors response: We agree that the introduction could be shortened, and thank the reviewer for their suggestions on how to achieve this. We have shortened the introduction where we feel it is appropriate to do so. This has included removing much of the detail on the individual tests, and we have moved the descriptions of the subtests to a new table. Additional references signpost readers to where they can find details on test format and marking structure for those who are interested. We feel these changes have improved the readability of the paper, while maintaining the necessary detail for readers to understand this paper.

Reviewer 1: More synthesis will help the reader to appreciate the lack of knowledge which motivated the study (" Currently, no evidence relates to how BMAT or UCAT scores relate to performance on PACES."), the goal and objectives of the study (page 8).

Authors's response: We had included most of the relevant evidence relating to the predictive validity of BMAT and UCAT in the original draft. However, we have expanded on the discussion of these points, which hopefully will help reinforce the motivations of this study, as helpfully suggested by the reviewer.

Reviewer 1: A second point - a suggestion - is a clear statement on the theoretical perspective adopted by the authors. Selection and prediction are complex processes, based on assessment (see eg. Patterson, F., et al (2016). How effective are selection methods in medical education? A systematic review. *Medical education*, 50(1), 36–60.). Assessment is inherently based on values and theoretic assumptions must be declared, because the constructs that underly BMAT, UCAT are meaningful only in a theoretical framework. As far I can understand, the authors adopted a cognitive quantitative approach. A licit choice, that is not the only possible one. It could be confusing to mix up socio-cultural determinants or "personality traits such as conscientiousness ... inter-personal functioning ... social intelligence" (page 19) that belong to other theoretical backgrounds. I think it would be fair to declare that the study considers only some of the cognitive elements of the very complex process of the development of a "good doctor".

Authors response: We accept that our theoretical framework should be made explicit. We also completely agree that 'cognitive abilities' (as traditionally, narrowly defined) are only one element important to the development of highly competent clinicians. We now declare this as suggested by the reviewer. Indeed, this is a point at least one of the authors has made in separate articles and studies. Our theoretical framework is drawn from the "individual differences" field, and, specifically the related theory related to the psychometrics of "intelligence", and how these can be applied to the

interpretation of cognitive test scores (see Flanagan et al. 2013, now included as a reference). This is now made clear in the background section. Moreover, the relevant individual differences literature sometimes highlights the overlap between concepts traditionally related to “cognitive ability” and “non-cognitive abilities”, such as the potential link between verbal reasoning and “social intelligence”. This is mentioned in the discussion section with appropriate supporting references. Nevertheless, for clarity, in the revised paper, we are now clear that such personal qualities, such as “conscientiousness” are viewed, within our chosen theoretical framework, as separate, but mediating factors, between cognitive ability and actual academic performance.

Reviewer 1: Finally, another possible use of the results of the study could be to inform the educational process. Beyond using the scores to select students, a faculty could use the score to personalize some interventions to achieve the best possible outcome for every student.

Authors' response: This is a good point, and we have added a paragraph into the discussion reflecting this alternative use of the results.

Reviewer: 2

Dr. Reinaldo Bestetti, University of Sao Paulo Faculty of Medicine of Ribeirao Preto

Comments to the Author:

In this paper, the authors have evaluated the role of Biomedical Admission test (BAMT) and University Clinical Admission Test (UCAT) in the prediction of academic achievement at the Membership of the Royal College of Physicians (MRCP) at the first attempt. The MRCP, a postgraduate assessment, is comprised of part I (knowledge-based), part II (applied knowledge), and part III -Practice Assessment of Clinical examination Skills (PACES), which is devoted to the assessment of aptitude and skills. A particular attention was paid to PACES.

The authors studied 3045 subjects, and observed that the aptitude and skills as well as scientific knowledge and applications subtest of the BMAT predicted passing in MRCP I and II at the first attempt. However, aptitude and skills were the only subtests to predict passing in MRCP-PACES at the first attempt. Regarding UCAT, quantitative reasoning and verbal reasoning predicted passing in the MCRP part I, and verbal reasoning (the ability to problem-solving) predicted passing MCRP Part II and MCRP PACES. Importantly, these results were obtained independently of previous academic attainment at high school. Thus, they concluded that “selectors may wish to consider placing particular weight on scales assessing these attributes “[“thinking skills”i.e problem solving ability - aptitude and skills) or verbal reasoning] if they wish to select applicants likely to become more competent clinicians.”

GENERAL COMMENTS. The selection process of students to enter a medical school is a difficult task. The future doctor should acquire cognitive abilities, procedural skills, and compassion attitudes, which are difficult to predict at the entrance examination test. Therefore, it was reassuring to take a look at this paper in which we can see that the assessment of aptitude and skills and verbal reasoning, ultimately the capacity of problem-solving, are predictive of academic achievement at a postgraduate medical examination.

SPECIFIC COMMENTS. By and large, the paper is nicely written.

Authors' response: Thank you

Reviewer 2: I could suggest some changes on it; 1) I think that the Introduction section is too long, and should be shortened accordingly.

Authors' response: We agree with this, and have reduced the introduction. Please see response to Reviewer 1 for more details.

Reviewer 2: 2) In my view, the use of the term “thinking skills” is not appropriate. Therefore, I would suggest to the authors to keep the terms aptitude and skills instead;

Authors' response: Previous research has shown that aptitude and skills can be conceptualised as measuring 'Thinking skills', and this section of BMAT has subsequently been renamed to reflect this. As such, there are times, when referring to the BMAT as it currently stands or when discussing the underlying construct of BMAT section 1, where it is necessary to use the term 'thinking skills'. This is because 'Thinking Skills' is the actual label, given by the test developer, of that particular subtest of the BMAT. However, we have replaced the term *thinking skills* with *aptitude and skills* where we are explicitly referring to results from this study. We hope that the reviewer finds this reasonable.

Reviewer 2: 3) As I understood the situation, attitudes were not assessed in BAMT, UCAT, and MCRP. If this is true, I think that it is an important limitation because attitudes are paramount for a good doctor; therefore, a few comments on the Discussion section would be desirable;

Authors' response: In the context of this paper, this is correct (NB: the UCAT does contain a situational judgement test component, which in some sense assess 'attitudes', but we did not use the SJT in this paper as it was not introduced until 2013). We have added in a comment to the discussion to reflect this.

Reviewer 2: 4) The Discussion section is also too long, and authors should focus on the explanation given to the results obtained. Therefore, I strongly suggest that the Discussion section be shortened.

Authors' response: We have removed some detail from the discussion, focussing more explicitly on the results obtained in this study. We have made edits to shorten the discussion more generally too.

VERSION 2 – REVIEW

REVIEWER	Fabrizio Consorti University of Rome La Sapienza
REVIEW RETURNED	13-Dec-2021
GENERAL COMMENTS	I'm fully satisfied with the changes the authors did
REVIEWER	Reinaldo Bestetti University of Sao Paulo Faculty of Medicine of Ribeirao Preto, Medicine
REVIEW RETURNED	07-Dec-2021
GENERAL COMMENTS	I think that everything is now OK with the revised version of the manuscript.